# A Honey Bee In-and-Out Counting Method Based on Multiple Object Tracking Algorithm

**DOI:** 10.3390/insects15120974

**Published:** 2024-12-06

**Authors:** Chaokai Lei, Yuntao Lu, Zhiyuan Xing, Jie Zhang, Shijuan Li, Wei Wu, Shengping Liu

**Affiliations:** Key Laboratory of Agricultural Blockchain Application, Agricultural Information Institute, Chinese Academy of Agricultural Sciences, Ministry of Agriculture and Rural Affairs, Beijing 100081, China; 13520493050@163.com (C.L.); 82101221354@caas.cn (Y.L.); 15558053226@163.com (Z.X.); zhangjie@caas.cn (J.Z.); lishijuan@caas.cn (S.L.)

**Keywords:** honey bee, in-and-out activity, object detection, multiple object tracking, counting method

## Abstract

The honey bee (*Apis mellifera*) is of great significance to both the ecological environment and human society. Systematic monitoring of honey bee activities is a useful method for determining the colony condition and can facilitate pre-emptive measures against detrimental conditions. It is relatively easy to digitally monitor and analyze the incoming and outgoing actions of bees in current research. Cameras within beehives were used to collect the activity data of bees. Our work mainly involved the application and evaluation of algorithms and methods. Object detection algorithms and multiple object tracking algorithms were used to track the honey bees’ incoming and outgoing actions. Based on the object detection algorithm and the multiple object tracking algorithm, in-and-out counting methods (the single-line method and the box method) were designed. After three levels of evaluation, the results showed that the in-and-out counting model consists of YOLOv8m, OC-SORT, and the box method, which achieved the best performance, and the model can be used to analyze the activity and colony condition.

## 1. Introduction

The honey bee (*Apis mellifera*) is of great significance to both the ecological environment and human society, providing bee products and making a significant contribution to the pollination of crops [1]. Beekeeping is an essential component of the agricultural industry, and precision beekeeping (PB) [2,3] is a field of agriculture that aims to protect bees, support beekeepers, and optimize apiary production through digital infrastructure. The systematic monitoring of honey bee activities also belongs to precision beekeeping, is instrumental in assessing the bee colony conditions, and can facilitate pre-emptive measures against detrimental conditions [4,5]. In the current research, monitoring and analyzing bees’ incoming and outgoing actions digitally is relatively easy. The most intuitive observation is the incoming and outgoing actions at the entrance of the beehive, because it is the essential route for exchanging food and energy between the bee colony and the outside world [6]. The activity of bees incoming and outgoing the hive can reflect the internal conditions of the colony, but it requires the accurate counting of the number of bees entering and leaving the hive. Ngo, et al. [7] designed a metric from incoming and outgoing counts to measure the daily loss of bees in order to assess the effects of the external environment.

Counting the number of in-and-out bees manually is time-consuming, posing challenges for large-scale data collection. The advancement in sensor technology has led to the adoption of automated monitoring methods over manual counting. Presently, automated monitoring methods fall into two categories: sensor-based and image/video-based. Infrared sensor and radio frequency identification (RFID) are two widely used sensor-based methods [8]. Although infrared sensors can monitor the in-and-out activity relatively accurately, the results are easily affected by the external environment, such as air humidity, and the bees’ own behaviors [9]. RFID necessitates affixing tags to bees’ backs, leading to tag wastage and potential harm to the bees [10,11]. In contrast to the aforementioned sensor-based monitoring, image/video-based techniques provide more direct monitoring of bee behavior and are less influenced by colony dynamics. Computer vision technology is used for processing images and videos, facilitating efficient and precise object detection and tracking. Conventional computer vision tasks depend on analytical, statistical methods, or classical signal processing [12]. Chiron, et al. [13] presented a stereo-vision-based system and proposed a detect-before-track approach that employs two innovating methods, as follows: hybrid segmentation and tuned 3D multiple object tracking based on the Kalman filter and Global Nearest Neighbor. Ngo, et al. [7] employed background subtraction for segmenting moving honey bees, and it effectively incorporated the Kalman filter and Hungarian algorithm to track multiple bees. Although conventional computer vision has more advantages in processing speed, it relies on manually annotated features, which are more time-consuming. In addition, it is susceptible to background change, and robustness and scalability are also weaknesses [14]. These drawbacks make it challenging to effectively identify and count bees in multiple different hives in practical scenarios. Considering the poor adaptability of the conventional methods, there is a necessity for a more efficient approach to detect and track honey bees.

In recent years, machine learning has been widely utilized across various domains, particularly in computer vision [15,16]. Deep learning represents a subset of machine learning algorithms designed to extract high-level features from input data. Since 2012, computer vision based on deep learning has dominated the domain due to substantially better performance compared with the conventional computer vision methods [17]. Object detection is an essential computer vision task rooted in deep learning [18], identifying visual objects like humans, animals, or cars. This technology finds extensive application in agriculture, including underwater fish detection [19], bird identification in forest environments [20], aphid detection [21], and bee detection [22,23,24,25].

MOT (multiple object tracking) locates multiple objects, maintains their identities, and yields their trajectories based on the features extracted by object detection [26]. Studies of bee tracking focus on bee in-and-out activity [27,28,29], pollen detection [30], and other behavioral tracking [31,32,33,34]. In counting in-and-out activity, related works show the designed observed box [7] and specific gate [27], which can improve tracking and counting accuracy. However, the special gate design can disrupt normal bee activities and potentially cause blockages, limiting its widespread deployment in practical beekeeping scenarios. Information on the related studies is shown in Table 1.

It is essential to develop an in-and-out counting model based on object detection and MOT. The current studies focus more on the applicability of bee detection and in-and-out tracking. Fewer metrics have been analyzed for tracking and counting bees, and they have failed to discuss the quantitative indicators of the models. In our study, MOT and counting metrics were introduced, which provided a more precise evaluation of the model’s performance. In order to achieve accurate tracking and counting of bee in-and-out activity and better grasp bee colony conditions, this study aims at the following: (1) to develop a precise method for counting incoming and outgoing bees; and (2) to evaluate the performance of multiple algorithms in the in-and-out activity counting model.

## 2. Materials and Methods

### 2.1. Experiment Equipment

This study utilized a smart beehive for video data collection. The equipment consisted of a camera, a host, and a beehive. The camera (Huiboshi model X20, Shenzhen Huiboshi Technology, Shenzhen, China) was positioned approximately 30 cm above the entrance of the beehive to provide continuous monitoring of the bees’ in-and-out activity. The parameters of the smart beehive are shown in Appendix A. The smart beehive is powered by PoE (Power over Ethernet), and data are transmitted to the cloud through Wi-Fi. We analyzed the video data stored in the cloud to count the number of incoming and outgoing bees. The smart beehive and monitoring system are shown in Figure 1.

### 2.2. Data Acqiusition

The honey bees (*Apis mellifera*) monitored in this study are Italian honey bees, a subspecies of Apis mellifera. The two honey bee colonies were raised in an apiary located in an experimental bases in the Haidian District (June 2022 to January 2023) and Changping District (March 2023 to July 2023), Beijing, China. Each colony had three frames of honeycombs in March 2023, and the new frames were added according to the number of honey bees. The smart beehives were used to continuously collect videos of the entrance of two beehives during these two periods. The video data are 80TB and were used to build the dataset.

### 2.3. Datasets and Image Labeling

The dataset in this experiment was divided into three parts, the training dataset and validation dataset for training of object detection (OD-BEE), the test dataset of MOT (MOT-BEE), and the test dataset of the in-and-out counting algorithm (IOC-BEE). According to different sources of OD-BEE, this dataset can be divided into three parts, and Table 2 shows information on the OD-BEE dataset. OD-BEE 3 includes the images obtained by frame extraction and cropping from the video data. OD-BEE includes bees under various densities and light intensities and comprises the training set (83%) and validation set (17%). OD-BEE was annotated using labelImg (https://github.com/HumanSignal/labelImg, accessed on 1 April 2023). MOT-BEE consists of eight videos of different durations, and the videos were labeled using Darklabel (https://github.com/darkpgmr/DarkLabel, accessed on 15 June 2023). IOC-BEE consists of eight videos, including different in-and-out counts and environmental conditions. Figure 2 shows the samples of the mentioned datasets, and more details of these datasets can be found in Appendix A. 

### 2.4. Model

As illustrated in Figure 3, the bee in-and-out activity counting model consists of the object detection algorithm, MOT algorithm, and in-and-out activity counting method. The video stream is input into the counting model, and the output is the incoming and outgoing counting results.

#### 2.4.1. Object Detection

YOLO (You Only Look Once) [35], as a one-stage object detection algorithm, offers advantages such as fast speed, simplicity, and efficiency. YOLO is known for being a real-time system that predicts bounding boxes and probabilities for each grid cell by dividing images into a grid. YOLOv8 [36] is an improved model released by Ultralytics and introduces new features and improvements to boost performance and flexibility further. The structure of YOLOv8 and the parameters of different YOLOv8 versions are shown in Figure 4. YOLOv8 comprises a backbone network, a neck network, and a prediction output network. The backbone leverages convolutional operations to extract the characteristics of various scales from RGB (Red Green Blue). The neck network’s function is to perform feature fusion and enhancement. The head network is the decision-making part of the object detection model used to achieve the final detection results. The loss function of this object detection algorithm can be found in the research of Yang, et al. [37]. The parameters of YOLOv8, from small to large, are as follows: YOLOv8n, YOLOv8s, YOLOv8m, YOLOv8l, and YOLOv8x. All versions of YOLOv8 were tested.

#### 2.4.2. MOT

MOT can be grouped into two sets, Detection-Free Tracking (DFT) and Detection-Based Tracking (DBT), depending on how the objects are initialized [26]. DFT requires manual initialization of a fixed number of objects in the first frame, then localizes these objects in subsequent frames. In video data, honey bees continuously appear and disappear, and long-term manual annotation is required, which is time-consuming. DBT is more popular because new objects are discovered and disappearing objects are terminated automatically.

Simple online and real-time tracking (SORT) [38] is a DBT algorithm based on the Hungarian algorithm [39] and Kalman filtering (KF) [40]. Many MOT algorithms have been enhanced using SORT as a foundation. The geometry of each object’s bounding box is estimated by predicting its new location in the current frame. The assignment cost matrix is then computed as the intersection-over-union (IOU) distance between each detection and all predicted bounding boxes from the existing targets. The assignment is solved optimally using the Hungarian algorithm. Furthermore, a minimum IoU threshold is enforced to discard assignments with an overlap below IoUmin. Any newly detected target will progress to the subsequent matching stage, while matched targets utilize Kalman filtering. The structure of SORT is shown in Figure 5. Based on the object detection models, the commonly used object tracking models are as follows: ByteTRACK [41], BotSORT [42], StrongSORT [43], OC-SORT [44], and Deep OC-SORT [45]. In addition, details of BotSORT, StrongSORT, ByteTRACK, OC-SORT, and Deep OC-SORT are shown in Appendix A, and these algorithms were used in our study.

#### 2.4.3. In-and-Out Counting Method

The detection and tracking of bees are the prerequisites for counting the in-and-out activity of the bees. In the MOT of honey bees, the shape difference between the different bees is slight, leading to many errors in bee tracking, such as Identity switch (IDSW). IDSW is a common error where the identification of one bee is mistakenly assigned to another. In both vehicle tracking [46] and pedestrian tracking [47], the counting method is achieved by a virtual detecting line at a specific location, which can reduce the impact of IDSW. This method involves storing the ID and position of the tracked objects. For instance, multiple objects were tracked at frame t. When an object crosses the virtual test line at frame t + 1, the counter is incremented by 1. The counting method based on the detection line only processes the positional information of the two consecutive frames of the specific detected bees, which greatly reduces the impact of IDSW.

(1)The single-line method

A virtual detection line is placed in front of the beehive entrance, with its length matching the width of the hive entrance, as depicted in Figure 6a. This approach is termed the single-line method and is utilized to track the in-and-out activity of bees crossing the detection line. The determination of incoming or outgoing actions is based on the vector angle. In Figure 6b,c point M is the object’s center point in the previous frame, and point N is the object’s center point in the subsequent frame. P and Q are the two endpoints of the detection line. Here, θ is the vector angle between MN and PQ. If θ > 0, the incoming counter will add 1. Conversely, if θ < 0, the outgoing counter will add 1.
(1)$$\theta =\tan\;(\vec{AB},\vec{CD})$$

(2)The box method

Based on the position of the camera in relation to the entrance of the bee colony and the bees’ movements, certain honey bees entered or left the beehive from below the detection line. They are, therefore, not detected, resulting in counting errors because they did not cross the detection line. The detection line being situated above the entrance of the beehive does not resolve these counting errors. Based on the line’s position of the single-line method, a new detection line was placed under the entrance of the beehive, and two detection lines were added beside the entrance, which was named the box method, as shown in Figure 6d. However, the box method presented another challenge (bee#3 in Figure 6), wherein some bees merely traverse the detection box without entering the hive directly. This problem can be solved by counting algorithm design.

Figure 7 shows the box method algorithmic process. Firstly, multiple variables were initialized. The In-count was used to record the number of incoming honey bees, and the Out-count was used to record the number of outgoing honey bees. The video is divided into multiple frames by their number. The incoming and outgoing actions were judged through the method shown in Figure 6b,c. If a honey bee crossing the detection line is judged as an incoming bee, the In-count will add 1. If a honey bee crossing the detection line is judged as an outgoing bee, the Out-count will add 1. If a bee enters the detection box and exits immediately, the In-count and Out-count will both add 1, which results in an error. In addition, the variable (Already-counted) is a double-ended queue and can solve the mentioned error. Already-counted is used to record the ID of the honey bee that has entered the detection box. If a bee has exited the detection box but its ID has been recorded in Already-counted, the variable will remove the ID, and the In-count will minus 1. After accessing all of the objects in a frame, the next frame will access the data until it finishes all of the frames of a video. Moreover, it will output the number of incoming and outgoing bees.

### 2.5. Model Evaluation

The performance of object detection, MOT, and in-and-out counting algorithms were evaluated in this study. Precision, Recall, and mAP were used to evaluate the object detection algorithms [48]. MOTA (multiple object tracking accuracy) and IDF1 (Identity F1 Score) constitute critical metrics for MOT. Precision, Recall, and the F1 score were used to evaluate the in-and-out counting methods. Moreover, FPS (frames per second) denotes the speed at which the detection model identifies targets within an image, and it is also an important evaluation metric. FPS can also be used to evaluate the speed of tracking and counting.

#### 2.5.1. Object Detection

mAP@50 denotes the mAP value when the confidence threshold is set at 0.5. mAP@50:5:95 represents the mean average precision within a confidence interval ranging from 0.5 to 0.95 using a step size of 0.05. Typically, a higher mAP@50:5:95 value signifies a superior model performance. Moreover, frames per second (FPS) denotes the speed at which the detection model identifies targets within an image. The processing speeds differ based on the device setup, and FPS serves as a metric for comparing the processing rates of various models under identical conditions. The discussion further encompasses the processing speed associated with tracking and counting tasks.

#### 2.5.2. MOT

MOTA captures all errors in object configuration resulting from misses and mismatches across all frames [26]. IDF1 represents the ratio of accurately identified detections to the average number of ground truth and computed detections [49].
(2)$$MOTA=1-\frac{FN+FP+IDSW}{GT}$$
(3)$$IDF1=\frac{2}{\frac{1}{IDP}+\frac{1}{IDR}}=\frac{IDTP}{IDTP+\frac{IDFP}{2}+\frac{IDFN}{2}}$$

In (2), IDSW is used to count the number of ID switches during the tracking process. IDSW is an important variable for deriving other MOT metrics, and it also introduces the number of true labels (ground truth, GT). FN (false negative) is the number of undetected ground-truth bounding boxes and FP (false positive) is the number of incorrect detections of nonexistent objects or a misplaced detection of existing boxes. In (3), IDTP (Identity true positive) is the number of identities of detections correctly matched with the ground truth in the trajectories. IDFN (Identity false negative) is the number of identities of detections incorrectly matched with the ground truth in the trajectories, and IDFP (Identity false positive) is the number of undetected ground-truth bounding boxes in the trajectories [50]. Christoph Heindl’s codes (https://github.com/cheind/py-motmetrics, accessed on 15 June 2023) were used to calculate the MOT metrics.

#### 2.5.3. In-and-Out Activity Counting Algorithm

While the model count may align with the manual count, it is highly probable that missing counts and miscounts can arise concurrently. Therefore, the error rate is not an appropriate evaluation metric, so Precision, Recall, and the F1 score were used to evaluate the methods.
(4)$$Precision=\frac{TP}{TP+FP}$$
(5)$$Recall=\frac{TP}{TP+FN}$$
(6)$$F1score=\frac{2\times Precision\times Recall}{Precision+Recall}$$

In (4) and (5), the actual incoming and outgoing counts that are correctly and recorded are FP, the targets that meet the conditions above but are not recorded as FN, and the targets that do not enter or leave but are recorded as FP. The F1 score is the reconciled average of the two metrics. The Precision, Recall, and F1score of the incoming and outgoing actions are calculated separately.

## 3. Results

### 3.1. Results of Object Detection Algorithms

Various versions of YOLOv8 were trained (Table 3) and mAP@50, mAP@50:5:95 and Speed are metrics to evaluate trained various versions of YOLOv8. mAP@50 can’t distinguish their performance and mAP@50:5:95 is a more accurate metric. YOLOv8m achieved the highest mAP@50:5:95 of 77.64%, with a relatively high speed. The trained YOLOv8m was assessed across diverse conditions involving scenarios with varying bee populations and different environmental light intensities (Figure 8). YOLOv8m demonstrated reliable object detection capabilities throughout these assessments. Trained YOLOv8m was used as the detector to test the MOT algorithms.

### 3.2. Results of MOT Algorithms

To improve tracking performance, five MOT algorithms based on YOLOv8m were evaluated. According to Table 4, OC-SORT is the most suitable algorithm for our work, and demonstrates the highest performance compared to the other MOT algorithms, maintaining the highest MOTA of 76.09% and an IDF1 of 67.83%, with a speed of 21.99 FPS. ByteTRACK achieves 66.92% and 68.62% in IDF1 and MOTA, respectively, and Deep OC-SORT achieves 59.52% and 74.30% in IDF1 and MOTA, respectively. The performance evaluation of BotSORT and StrongSORT yielded subpar results, and they will not be used in in-and-out activity counting either. Obviously, OC-SORT, with the highest IDF1 and MOTA, is more suitable for bee tracking.

The performance of the MOT algorithm in a low light intensity environment needs to be tested, as well as OC-SORT based on YOLOv8m-processed video data from early morning, assessing the detection and tracking outcomes as the light intensity varied. Since the light intensity cannot be directly tested, we employed the average image brightness as its surrogate measure. The calculation method for the average brightness of the image is to add up the brightness values of all pixels in the image and then divide it by the total number of pixels to obtain the average brightness, with a maximum of 255 and a minimum of 0. In Figure 9a,b, the bee’s target boxes are currently lost. In (c) and (d), the bee can be tracked stably, and the ID remains stable. When the average brightness in the scene is higher than 59.09, the model can achieve stable tracking.

### 3.3. Results of Counting Methods

ByteTRACK, OC-SORT, and Deep OC-SORT performed well in MOT-BEE, and they were tested with the in-and-out activity counting test dataset. Table 5 shows the Precision, Recall, and F1 of the incoming and outgoing action counting results using the single-line method. F1_in_ and F1_out_ of OC-SORT achieved 73.78% and 74.40%, respectivley. Deep OC-SORT performed worse than OC-SORT. The single-line method based on ByteTRACK and OC-SORT attained a processing speed of 20 FPS. Although the F1_out_ of ByteTRACK is higher than that of OC-SORT, its F1_out_ is much lower than that of the F1_out_ of OC-SORT.

The box method was tested based on ByteTRACK, OC-SORT, and Deep OC-SORT, and the test results are shown in Table 6. Compared to the single-line method, the box method achieved higher Precision, Recall, and F1 score for counting the incoming and outgoing bees from the hive entrance, without a significant reduction in operational speed. According to manual observations, the box method solved errors that detected bees that crossed the detection line without entering the beehive and bees that entered or exited the beehive without crossing the detection line in the single-line method. The box method based on OC-SORT demonstrated a strong performance, achieving F1_in_ of 91.49% and F1_out_ of 89.08%. According to the results, the in-and-out counting model, which consists of YOLOv8m, OC-SORT and the box method, performed well.

### 3.4. The Application of the Selected Model

The in-and-out counting model was used to count the incoming and outgoing bees for specific dates. Three days in March and April 2023 were chosen to display the counts of incoming and outgoing bees per hour (Figure 10). Comparing horizontally, the in-and-out activity reveals a clearer circadian rhythm, with more counts at noon and fewer in the morning and evening [7].

## 4. Discussion

### 4.1. Comparison of MOT Algorithms

Our study has incorporated MOT metrics to evaluate the MOT algorithms’ performance, enhancing the credibility and applicability of our approach [50]. In MOT, the ByteTRACK algorithm emphasizes the stability of the tracking trajectories. The Byte module in ByteTRACK is designed to retain low-score detection frames, ensuring the matching of all detected targets, leading to minimizing IDSW [42]. The lower dropout rate of the Byte model will lead to a significant increase in FN, so the MOTA does not increase. Deep OC-SORT is incorporated into the dynamic appearance module, and the advantage is that more targets can be detected, which concurrently leads to an escalation in IDSW and a decline in MOTA and IDF1. In the bee tracking field, there are minimal differences in the appearance of the different bees, and even the specific bee’s ID may be changed in the next frame. The appearance-based tracking algorithm proves less effective when dealing with targets sharing similar appearances or when facing occlusion [46], therefore, Deep OC-SORT does not show enhancement compared to OC-SORT. Additionally, the process of extracting the targets’ appearance features in Deep OC-SORT results in a longer runtime. Prioritizing the bee’s motion over appearance in tracking models is crucial within bee tracking. OC-SORT, centered on the motion of the target, can better adapt to honey bee tracking. The counting results are consistent with the tracking results.

### 4.2. Distance from the Detection Line to the Beehive Entrance

The distance from the detection line to the beehive entrance also influences the performance of the counting method. Four videos were chosen from IOC-BEE to find the proper distance. The distance of the single-line method based on YOLOv8m and OC-SORT algorithms was tested. Multiple distances were set from the detection line to the beehive entrance, which were 2, 1, 1/2, and 1/4 times the bee body length, and the minimum was close to the hive entrance (Figure 11). Table 7 demonstrates that setting the distance results in Precision_out_ and Recall_out_ of 0. The outgoing action was entirely disregarded, due to bees appearing above the detection line before being recognized. When the detection line distance is set to two, the performance of the counting method is poor, and excessive distance causes a lot of errors. The distance from the detection line to the beehive entrance is 1/2 of the bee’s body length with F1_in_ of 82.41% and F1_out_ of 77.60%. Thus, the optimal distance for detecting in-and-out actions is half of the bee’s body length.

As the distance approaches 0, the detected bees’ action is from or to the upper edge of the beehive entrance. This position of the detection line can be utilized for the box method, so that the line can be fixed in this position whose distance is nearly 0 to the beehive entrance. The length of the left and right detection lines varies with the position of the topmost detection line. Similar to the testing rules of the single-line method, the box method only needs to adjust the position of the topmost detection line. Similar to the selection of detection line positions in the single-line method, multiple detection box positions were set (Figure 12). The 1/2 distance from the topmost detection line to beehive distance achieves F1_in_ of 96.51% and F1_out_ of 93.08%, and the proper distance is also 1/2 (Table 8).

### 4.3. Comparison of Counting Methods

The counting method can be grouped into two sets: recording the complete trajectory and our proposed method. In the first method, the trajectories of the honey bees are documented, and the vector from the object’s initial position to the final position is used to distinguish between the incoming and outgoing actions [7]. ID switch can cause trajectory interruption, so this method amplifies the impact of ID switch. Our proposed method relies solely on the position of frames before and after a bee crosses the detection line, thereby reducing the impact of ID switches on the counting results. Compared to previous studies, the method proposed in this research significantly enhances the accuracy of counting bee entries and exits at the hive entrance.

The box method compensates for the shortcomings of the single-line method, considering incoming and outgoing actions from all directions, including the other three directions. Additionally, it effectively addresses errors related to passing actions, making the box method superior to the single-line approach. Nonetheless, IDSW can influence the accuracy of bee counting when employing the box method. The switching of IDs among bees entering and exiting the detection box, even if infrequent, can affect count Precision and Recall. Despite the potential insignificance of IDSW’s influence theoretically, its increased likelihood of occurrence poses a threat to the accuracy of both the incoming and outgoing counts. The in-and-out counting model, which consists of YOLOv8m, OC-SORT, and the box method, is the proposed approach to counting the honey bees’ incoming and outgoing activity.

### 4.4. Further Work

The advantage of video-based monitoring methods is that they can intuitively capture different bee behavior at the entrance. We will create a heatmap depicting the movement of bees to achieve a visual representation of the active area, evaluate the activity and movement speed of the bees, and calculate the speed of the detected bees, which will help to predict the activity of the bees at the beehive entrance. Additionally, the performance of object detection and MOT algorithms is directly related to the effectiveness of the in-and-out counting model, and we will focus on the development and progression of these algorithms. The swift movement of the honey bees in the videos results in their appearance as dark shadows in specific frames. The shadow and swift movement might lead to lower metrics of MOT and poorer performance of the counting model. Enhancing the resolution and frame rate of the data from the monitoring equipment can help to mitigate these problems.

## 5. Conclusions

Over the past decade, computer vision technology has played an increasingly important role in insect monitoring and pest management. With the advancement of software and hardware, the cost of insect behavior research has been reduced, and related research is gradually becoming more specific. This work proposes a bee in-and-out counting model, which consists of object detection, MOT, and in-and-out counting algorithms. YOLOv8m performs the best with mAP@50:5:95 of 77.64%. OC-SORT, focusing on object motion, is well-suited for bee tracking. The box method, aimed at error minimization, is appropriate for bee in-and-out counting. Future work includes integrating this model with additional dimensional data to analyze honey bee activity and environmental impacts.

## Figures and Tables

**Figure 1 insects-15-00974-f001:**
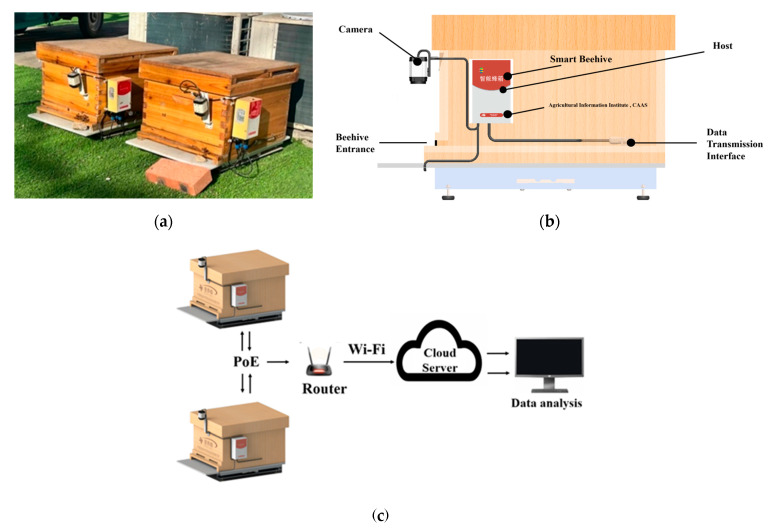
Smart beehive and monitoring system. (**a**) Deployment of smart beehive; (**b**) Smart beehive perspective drawing; (**c**) Monitoring system.

**Figure 2 insects-15-00974-f002:**
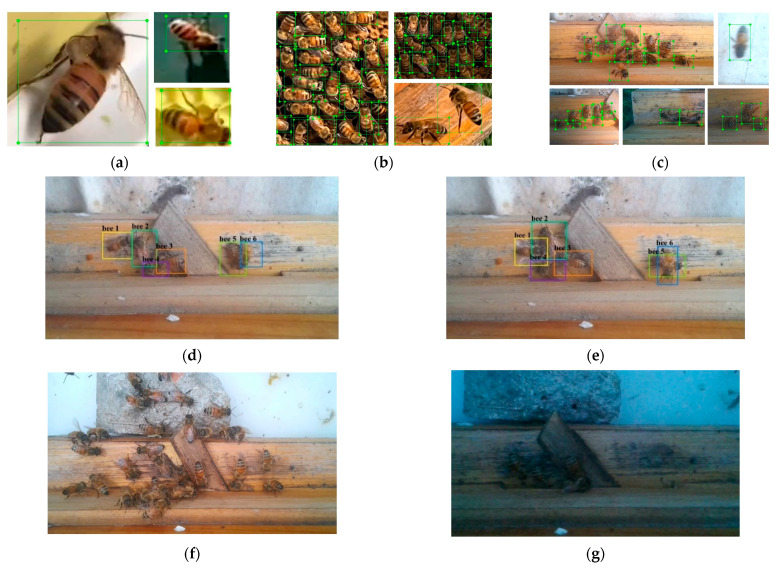
Samples of datasets. (**a**) OD-BEE 1; (**b**) OD-BEE 2; (**c**) OD-BEE 3; (**d**,**e**) MOT-BEE; (**f**,**g**) IOC-BEE. Figure 2a–c shows the annotation of OD-BEE, and Figure 2d,e shows the annotation of MOT-BEE.

**Figure 3 insects-15-00974-f003:**
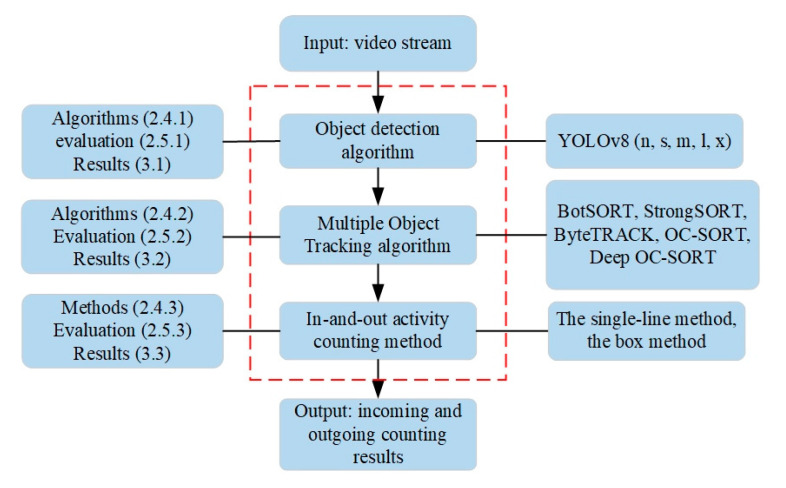
The structure of bee in-and-out activity counting model.

**Figure 4 insects-15-00974-f004:**
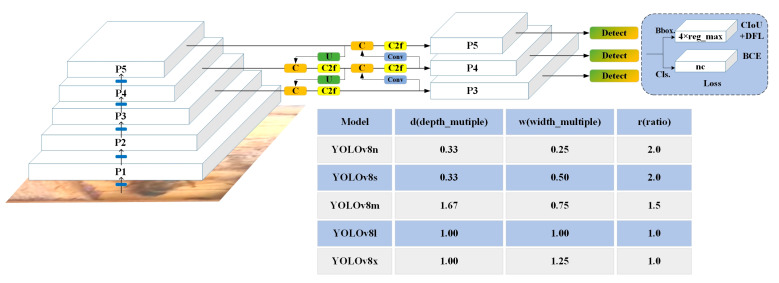
Detailed information of YOLOv8.

**Figure 5 insects-15-00974-f005:**
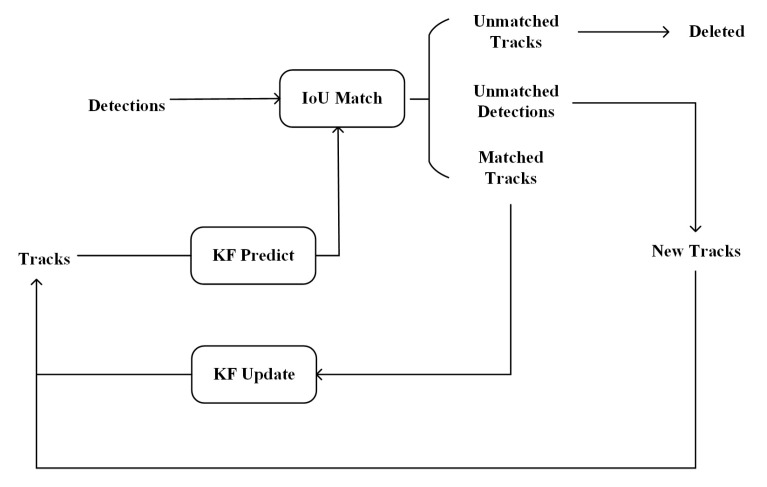
The structure of the SORT algorithm.

**Figure 6 insects-15-00974-f006:**
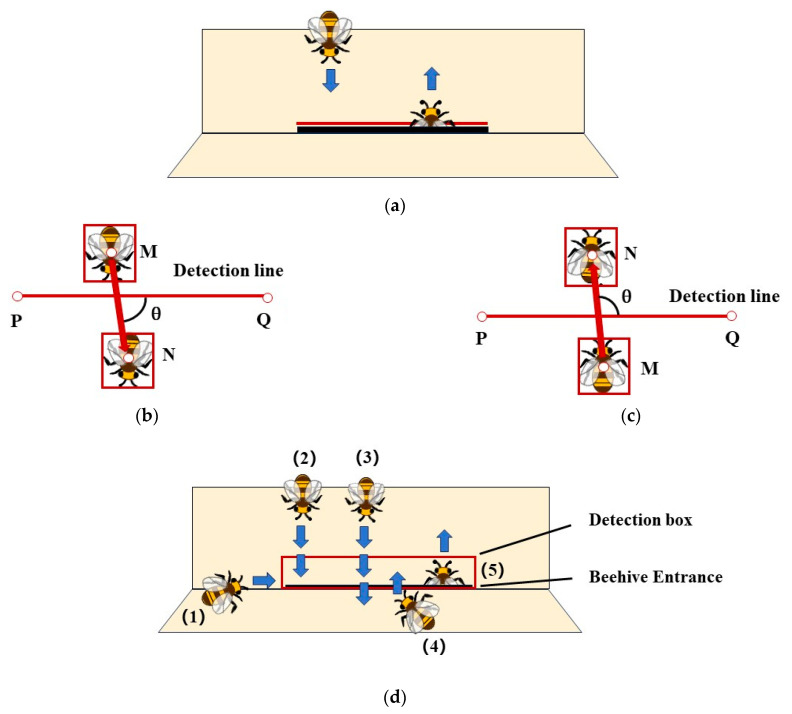
The in-and-out counting methods and the determination of incoming or outgoing actions. (**a**) The single-line method; (**b**) outgoing action; (**c**) incoming action; (**d**) the box method. The box method can detect honey bees from or to different directions (bee#1, 2, 4, 5 in Figure 6).

**Figure 7 insects-15-00974-f007:**
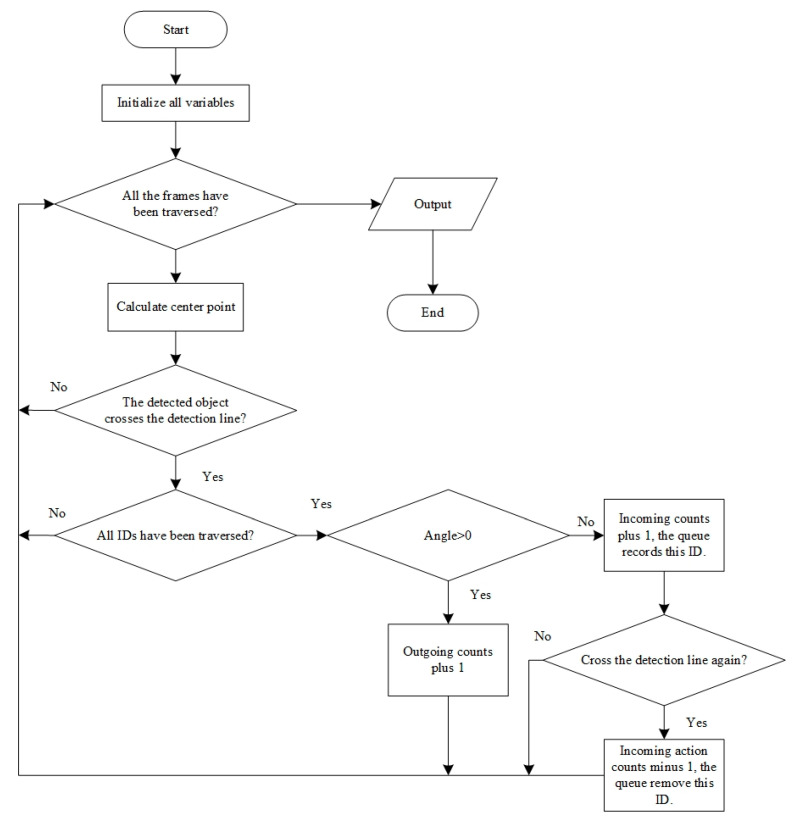
The box method algorithmic process.

**Figure 8 insects-15-00974-f008:**
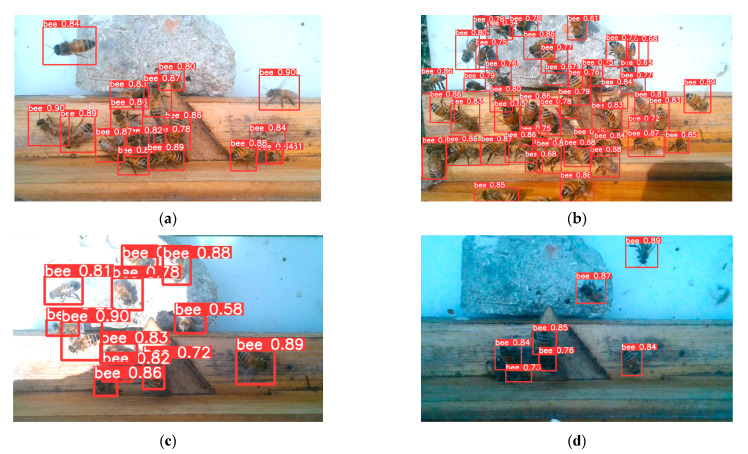
Diverse conditions involving scenarios with varying bee populations and different environmental light intensities. (**a**) Medium number of honey bees; (**b**) large number of honey bees; (**c**) high light intensity with shadow environment; (**d**) low light intensity.

**Figure 9 insects-15-00974-f009:**
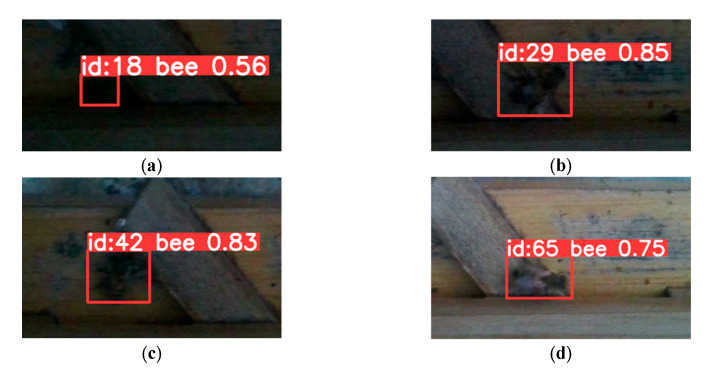
Influence of light intensity on detection and tracking. (**a**) 42.46; (**b**) 52.02; (**c**) 59.09; (**d**) 106.27.

**Figure 10 insects-15-00974-f010:**
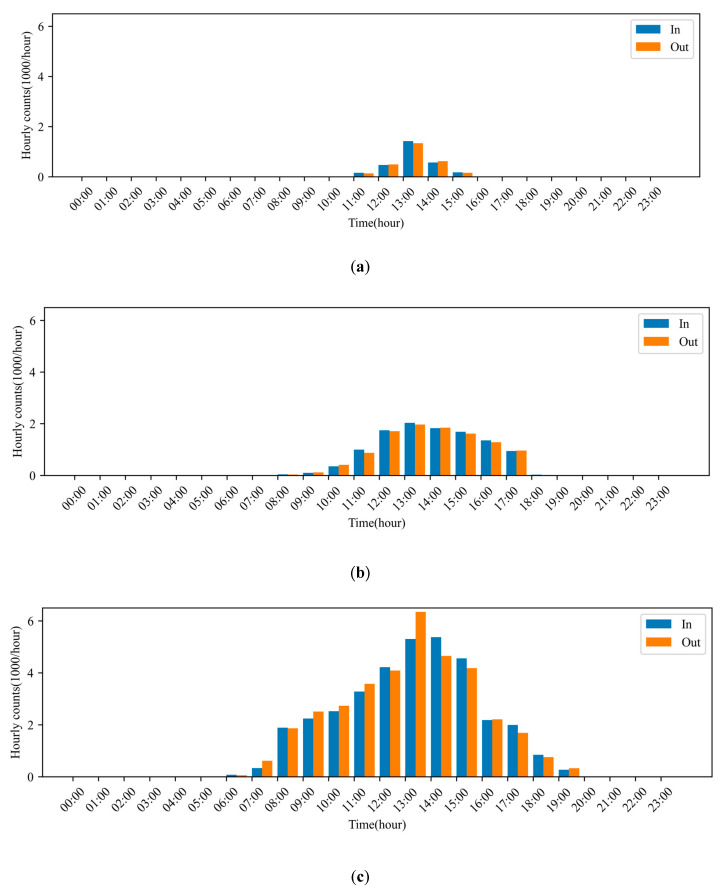
Hourly counts of incoming and outgoing actions. (**a**) 15 March 2023; (**b**) 7 April 2023; (**c**) 17 April 2023.

**Figure 11 insects-15-00974-f011:**
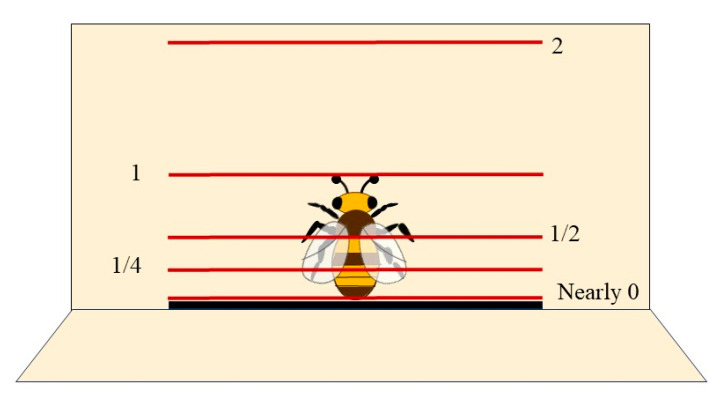
Different distances between the detection line and beehive entrance.

**Figure 12 insects-15-00974-f012:**
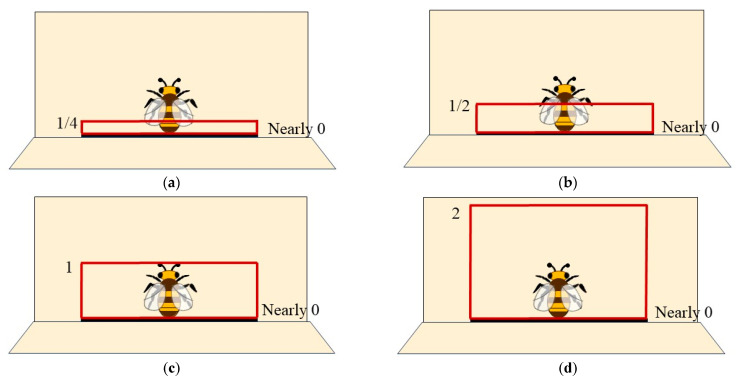
Different distances from the topmost detection to the beehive entrance. (**a**)1/4; (**b**)1/2; (**c**)1; (**d**) 2.

**Table 1 insects-15-00974-t001:** Detailed information on related works.

Authors	Works	Equipment and System	Methods
[13]	Detecting and tracking honey bees in 3D at the beehive entrance using stereo vision	Stereo vision cameras	Hybrid intensity depth segmentation, target extraction, and multi-target tracking in 3D
[7]	A real-time imaging system, model construction, and long-time monitoring	Beehive with observation box	Kalman filter and Hungarian algorithm
[27]	In-and-out monitoring system	Beehive with specific gate and camera	YOLOv4 and DeepSORT
[32]	Tracking individual honey bees among wildflower clusters with CV	Camera	Hybrid detection and tracking (HyDaT)
[30]	Tracking bees and bees with pollen and data analysis	Beehive, observation box, and other sensors	YOLOv3 tiny, Kalman filter, and Hungarian algorithm
[33]	Measuring honey bee traffic with BeePIV	Beehive and camera	BeePIV (dynamic background subtraction, particle image velocimetry)
[34]	Machine-learning-based bee recognition and tracking	Planar bee tracking system	YOLOv5 and SORT
[28]	Tracking model construction and a zone for mAP	Beehive and camera	YOLOv8m and ByteTRACK
[29]	Behavioral pattern recognition	Beehive and camera	YOLOv8m and ByteTRACK

**Table 2 insects-15-00974-t002:** Detailed information of OD-BEE dataset.

Content	OD-BEE 1	OD-BEE 2	OD-BEE 3
Source	Dataset published by the related study (https://www.kaggle.com/jenny18/honey-bee-annotated-images (accessed on 17 March 2023).)	Experimental staff with mobile phones	Video data from smart hive
Collection time	2 July 2018 to 1 September 2018	9 May 2023	25 June 2022 to 1 January 2023
Number of pictures	1033	100	2278
The average number per image	1	15	4
Behavior	Crawling	Crawling	Flying, crawling

**Table 3 insects-15-00974-t003:** Results of different YOLOv8 algorithms.

Metrics	YOLOv8n	YOLOv8s	YOLOv8m	YOLOv8l	YOLOv8x
Precision (%)	99.30	99.25	99.20	99.32	99.60
Recall (%)	98.08	98.35	98.59	98.67	98.61
mAP@50 (%)	99.40	99.45	99.37	99.45	99.45
mAP@50:5:95 (%)	75.72	76.99	**77.64**	76.99	77.29
Speed (FPS)	27. 52	**29.30**	21.29	14.48	10.93

**Table 4 insects-15-00974-t004:** Results of different MOT algorithms. The evaluation metrics such as IDF1, MOTA, and speed can evaluate the perfor-mance of MOT algorithms, higher values prove that the algorithm performed better.

Metrics	BotSORT	StrongSORT	ByteTRACK	OC-SORT	Deep OC-SORT
IDSW	4556	1356	578	1430	2891
IDF1 (%)	48.65	56.69	66.92	**67.83**	59.52
MOTA (%)	61.17	67.08	68.62	**76.09**	74.30
Speed (FPS)	10.11	17.17	**24.51**	21.99	17.49

**Table 5 insects-15-00974-t005:** Test results of the single-line method. Higher F1_in_, F1_out_ can prove the better performance of methods.

Metrics	ByteTRACK	OC-SORT	Deep OC-SORT
Precision_in_ (%)	65.03	69.15	49.09
Precision_out_ (%)	78.64	78.48	62.00
Recall_in_ (%)	71.54	79.07	72.31
Recall_out_ (%)	72.08	70.72	65.58
F1_in_ (%)	68.13	**73.78**	58.48
F1_out_ (%)	**75.21**	74.40	63.27
Speed (FPS)	**22.48**	21.67	17.23

**Table 6 insects-15-00974-t006:** Test results of the box method. Higher F1_in_, F1_out_ can prove the better performance of methods.

Metrics	ByteTRACK	OC-SORT	Deep OC-SORT
Precision_in_ (%)	83.64	94.29	80.43
Precision_out_ (%)	84.49	92.73	74.55
Recall_in_ (%)	88.46	88.85	86.92
Recall_out_ (%)	87.34	85.71	85.77
F1_in_ (%)	85.98	**91.49**	83.55
F1_out_ (%)	85.89	**89.08**	79.77
Speed (FPS)	**24.51**	21.99	17.49

**Table 7 insects-15-00974-t007:** Results of different YOLOv8 algorithms. Higher F1_in_, F1_out_ can prove the better distance from the detection line to the beehive entrance.

Distance	Nearly 0	1/4	1/2	1	2
Precision_in_ (%)	93.55	75.51	76.72	67.53	63.33
Precision_out_ (%)	0	82.02	84.52	81.08	63.33
Recall_in_ (%)	58.00	74.00	89.00	52.00	38.00
Recall_out_ (%)	0	73.74	71.72	60.61	19.20
F1_in_ (%)	71.61	74.75	**82.41**	58.76	47.50
F1_out_ (%)	/	**77.66**	77.60	69.37	29.47

**Table 8 insects-15-00974-t008:** Comparison of different distances from the topmost detection line to the beehive entrance. Higher F1_in_, F1_out_ can prove the better distance from the detection line to the beehive entrance.

Distance	1/4	1/2	1	2
Precision_in_ (%)	91.59	96.03	73.50	48.70
Precision_out_ (%)	93.07	90.38	78.43	48.84
Recall_in_ (%)	98.00	97.00	86.00	56.00
Recall_out_ (%)	94.95	95.95	80.81	42.42
F1_in_ (%)	94.69	**96.51**	79.26	52.10
F1_out_ (%)	**94.00**	93.08	79.60	45.40

## Data Availability

The algorithms and datasets used in this study can be found at the following address: http://github.com/lck981202/bee2 (accessed on 9 October 2023).

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
