# Peer review of "A Honey Bee In-and-Out Counting Method Based on Multiple Object Tracking Algorithm"

_insects, 2024, doi:10.3390/insects15120974_

Round 1

Reviewer 1 Report

Comments and Suggestions for Authors This research is mainly technical and applied, using a well-known machine-learning tool (specifically yolo8) to count bees entering and leaving the hive. The article's main contribution is a comparison of several different versions of the algorithm above + several comparisons between tracking paradigms of MOT (=multiple object detection) techniques.   The article is well written, but whether the result justifies an academic publication is unclear.   A significant improvement for the paper might happen if the suggested tool proposed in the paper will be used more significantly - for example, by a continuous comparison of the bee count along a continuous time window (e.g., 2 months) and not by describing only three short windows (in particular 3 days, see paragraph 3.19) including an extensive static analysis over the bee count (and the temperature, weather, light etc.)   If it is decided to publish the article, the following corrections must be made:   Insert the recent paper [1] to table 1:   [1] Rozenbaum, E., Shrot, T., Daltrophe, H., Kunya, Y., & Shafir, S. (2024). Machine learning-based bee recognition and tracking for advancing insect behavior research. Artificial Intelligence Review, 57(9), 245.‏   There is a need to explain the content of Figure 4 (or omit the figure).   Lines 207-210 :  there is only a high-level explanation of your chosen method. Please provide a detailed explanation using the illustration of Figure 4d (use also the number of the bees in the illustration)     Line 240, eq. (2): no detailed explanation of the content. Specifically, it is written: "FN and FP are the same as the metrics 243 in object detection." BUT it is NOT explained also in the object detection subsection.   line244: "The Identity F1 Score (IDF1) represents": do you mean IDSW in eq (2) ??? Where did you use this score?   Figure 3: it is a very important figure! It is recommended to insert references to the related subsections and tables of each phase, so that the structure of the paper and the meaning of each table will be more clear.   Line 317: "The in-and-out time range in a day will also be correspondingly extended.":  I didn't understand this sentence: Does it relate to previous works? or to your (future) work?    Minor comments:   Line 12: re-search-> research Line 82: MOT is first mentioned, but the meaning of the acronyms is NOT written (the abstract is NOT part of the paper!)

Author Response

Dear reviewer:

We would like to thank you for your comprehensive revisions and valuable comments on our submitted manuscript entitled “A honey bee in-and-out counting method based on multiple object tracking algorithm”. The responses are highlighted in red color, and the amended changes are highlighted in blue color. We look forward to working with you to move this manuscript closer to publication in Insects.

Comments 1: A significant improvement for the paper might happen if the suggested tool proposed in the paper will be used more significantly, for example, by a continuous comparison of the bee count along a continuous time window (e.g., 2 months) and not by describing only three short windows (in particular 3 days, see paragraph 3.19) including an extensive static analysis over the bee count (and the temperature, weather, light etc.)

Response 1: Thank you for your comment. In this manuscript, our work mainly focuses on the design and testing of the bee in-and-out counting model. Afterwards, we will analyze the condition of bee colony based on this method and other dimensions of data (temperature, weather and light etc.). We think this will be our future work. In 3.4, we just discuss the performance of methods and want to test whether the model can run stably.

Comments 2: If it is decided to publish the article, the following corrections must be made:   Insert the recent paper [1] to table 1: [1] Rozenbaum, E., Shrot, T., Daltrophe, H., Kunya, Y., & Shafir, S. (2024). Machine learning-based bee recognition and tracking for advancing insect behavior research. Artificial Intelligence Review, 57(9), 245.‏   

Response 2: Agree. We have inserted the recent paper in Table 1.

Comments 3: There is a need to explain the content of Figure 4 (or omit the figure).

Response 3: Agree. We have inserted the explanation of Figure 4 in lines 155-162.

Lines 155-162: YOLOv8 comprises a backbone network, a neck network and a prediction output network. The backbone leverages convolutional operations to extract characteristics of various scales from RGB (Red Green Blue). The neck network’s function is to perform feature fusion and enhancement. The head network is the decision-making part of the object detection model, to the final detection results. The loss function of this object detection algorithm can be found in the research of Yang et al [37]. Parameters of YOLOv8, from small to large, is as follows: YOLOv8n, YOLOv8s, YOLOv8m, YOLOv8l and YOLOv8x. All versions of YOLOv8 were tested.

Comments 4: Lines 207-210 :  there is only a high-level explanation of your chosen method. Please provide a detailed explanation using the illustration of Figure 4d (use also the number of the bees in the illustration)

Response 4: Agree. We have provided detailed explanation of Figure 6d in lines 215-219.

Lines 207-210 (now Lines 214-218): The box method can detect honey bees from or to different directions (bee#1, 2,4, 5 in Figure 6). However, the box method presented another challenge (bee#3 in Figure 6) wherein some bees merely traverse the detection box without entering the hive directly. We solved this problem from counting algorithm design.

Comments 5: Line 240, eq. (2): no detailed explanation of the content. Specifically, it is written: "FN and FP are the same as the metrics 243 in object detection." BUT it is NOT explained also in the object detection subsection.

Response 5: Agree. We have inserted the detailed explanation of eq. (2) in lines 253-255, and we deleted the confusing sentence.

Lines 258-260: FN (False negative) is the number of undetected ground-truth bounding boxes and FP (False positive) is the number of incorrect detections of nonexistent objects or a misplaced detection of existing boxes.

Comments 6: line244: "The Identity F1 Score (IDF1) represents": do you mean IDSW in eq (2) ??? Where did you use this score?

Response 6: Agree. We have inserted the introduction of IDF1 and its detailed definition. And IDF1 was used to evaluate MOT algorithms in Table 5.

Lines 253-255: IDF1 (Identity F1 Score) represents the ratio of accurately identified detections to the average number of ground truth and computed detections [49].

Lines 260-264: In (3), IDTP (Identity true positive) is matched on the overlapping part of trajectories that are matched together. IDFN (Identity false negative) and IDFP (Identity false positive) are the remaining ground truth and predicted objects respectively, from both non-overlapping sections of matched trajectories, and from the remaining trajectories that are not matched [50].

Comments 7: Figure 3: it is a very important figure! It is recommended to insert references to the related subsections and tables of each phase, so that the structure of the paper and the meaning of each table will be more clear.

Response 7: Agree. We inserted the reference of each phase.

Figure 3. The structure of bee in-and-out activity counting model.

Comments 8: Line 317: "The in-and-out time range in a day will also be correspondingly extended.": I didn't understand this sentence: Does it relate to previous works? or to your (future) work?

Response 8: Agree. We have deleted the sentence. This statement is not rigorous enough here, we will verify this in the future. And it will be our future work.

Comments 9: Minor comments: Line 12: re-search-> research Line 82: MOT is first mentioned, but the meaning of the acronyms is NOT written (the abstract is NOT part of the paper!)

Response9: Agree. We have modified the above issues.

Line 12: re-search-> research

Line 82 (now Line 85): MOT (Multiple Object Tracking)

Reviewer 2 Report

Comments and Suggestions for Authors

The paper introduces a novel approach by using the YOLOv8m model along with OC-SORT for tracking and counting bee activities, which is an innovative combination in the realm of bee activity monitoring. This application to bee tracking and the specific development of the "box method" for counting represents a significant contribution to precision beekeeping technologies.

The paper is well-organized,  however, I have the following concerns:

-  The statistical analysis presented in the model evaluation section (Section 2.5, Tables 4 and 5) could be strengthened. The paper would benefit from the inclusion of more robust statistical tests to justify the comparisons made between different models and to validate the significance of the observed differences. What are the standard deviation of the obtained data?

-  Object detection and tracking is something that can be done also without AI, doing images processing with very good results. Can the authors comment on this?

-  Considering Table 2, the number of pictures appears to be very low considering the recording time reported in the paper. With 1033, 199, and 2278 pictures and 30fps, the recording time is 34.43, 3.33, and 75.93 seconds, respectively. Am I missing something?

Considering the above points, it is challenging for me to consider it for publication.

Comments on the Quality of English Language

The English quality is a bit poor. Some sentences are not clear and are difficult to understand (e.g., Introduction and Section 2.2).

Author Response

Dear reviewer:

We would like to thank you for your comprehensive revisions and valuable comments on our submitted manuscript entitled “A honey bee in-and-out counting method based on multiple object tracking algorithm”. The responses are highlighted in red color, and the amended changes are highlighted in blue color. We look forward to working with you to move this manuscript closer to publication in Insects.

Comment 1: The statistical analysis presented in the model evaluation section (Section 2.5, Tables 4 and 5) could be strengthened. The paper would benefit from the inclusion of more robust statistical tests to justify the comparisons made between different models and to validate the significance of the observed differences. What are the standard deviation of the obtained data?

Response 1: Thank you for your comment. The way to analyze performance of the algorithms is to compared their metrics, because the metrics are clear. For example, in this table from Strongsort, the author directly compared metrics to confirm their performance, and all related papers do not use statistical analysis and standard deviation. It’s reasonable to compare different models using same train datasets and test dataset. And our sentences of evaluation are not clear, so we inserted the introduction of evaluation in lines 233-239.

Lines 236-242: There are the evaluations for object detection, MOT and in-and-out counting method separately. Precision, Recall, and mAP are used to evaluate object detection algorithms [48]. MOTA (Multiple Object Tracking Accuracy) and IDF1 (Identity F1 Score) constitutes critical metrics for MOT. Precision, Recall and F1 score is used to evaluate in-and-out counting methods. Moreover, FPS (frames per second) denotes the speed at which the detection model identifies targets within an image, and it’s also an important evaluation. FPS can also be used to evaluate the speed of tracking and counting.

Comment 2: Object detection and tracking is something that can be done also without AI, doing images processing with very good results. Can the authors comment on this?

Response 2: We have inserted the comment in our manuscript (Lines 71-75).

Lines 70-75: Although conventional computer vision has more advantages in processing speed, it relies on manually annotated features which is more time-consuming. And it’s susceptible to background change, and robustness and scalability are also weaknesses [14]. In the future, we will deploy the monitoring system in multiple beehives. Considering the poor adaptability of conventional methods, there is a necessity for a more efficient approach to detect and track honey bees.

Comment 3: Considering Table 2, the number of pictures appears to be very low considering the recording time reported in the paper. With 1033, 199, and 2278 pictures and 30fps, the recording time is 34.43, 3.33, and 75.93 seconds, respectively. Am I missing something?

Response 3: We collected a large amount of video data and obtained images through frame extraction and cropping. The images were used to train the YOLOv8. There is ‘The average number per image’ in Table 2. We labeled 11645 boxes to train the object detection algorithm. And there are several augmentation algorithms in YOLOv8 to enhance the richness of data. We have described the data sources in lines 118-120 and lines 126-128.

Lines 120-121: Smart beehives were used to continuously collect videos of the entrance of two beehives during these two periods. The video data is 80TB and was used to build the dataset.

Lines 125-129: According to different sources of OD-BEE, this dataset can be divided into three parts, and Table 2 shows the information of OD-BEE dataset. OD-BEE 3 is the images obtained by frame extraction and cropping from the video data. The OD-BEE includes bees under various densities and light intensities, and comprises training set (83%) and validation set (17%).

Comment 4: The English quality is a bit poor. Some sentences are not clear and are difficult to understand (e.g., Introduction and Section 2.2).

Response 4: Agree. We have improved the English quality, especially in Introduce and Section 2.2.

Reviewer 3 Report

Comments and Suggestions for Authors

General comments

Earlier researchers used manual counting method for counting the in-and-out numbers of honey bees on the hive entrance. This was a tedious method and there were chances of committing errors, especially when the numbers of in-and-out bees is very large, and the observer has to count the incoming numbers of pollen and nectar foragers separately. I am happy to see that the authors have given a better method of counting the bees. This method will help counting the bees precisely and accurately. Therefore, I find this article suitable for publication in ‘insects’, however, after some minor changes as suggested below under specific comments.

Specific comments

I differ a little with the repeated arguments of the authors regarding the usefulness of this method in determining the colony health, refer: lines 10, 19, 29 etc. I suggest the authors to making this statement little general, as ‘usefulness of method in determining the colony condition’.

Title: A better title can be, “A Multiple Object Tracking Algorithm Method for Counting the In-and-Out Numbers of Honey Bees”

Abstract:

Line 22: after the word inefficient, add two sentences, ‘Therefore, there was a need to devise some more efficient alternate method. With this purpose the present study was conducted’.

Line 25: What type of camera was used? Kindly write.

Introduction:

Lines 45-46: The number of bees in a colony— The condition of a colony

Line 46: outgoing counts---outgoing bees

Line 50: Counting bee in-and-out activity---Counting the numbers of in-and-out bees.

Line 90:  Kindly check the sentence.

Line 96: outgoing actions---outgoing bees

Lines 97-100: These are redundant, kindly delete.

Materials and Methods

Looks sound. Kindly make following changes:

Line 109: to obtain the number of bees counts of incoming and outgoing actions—change to--- to count the the number of incoming and outgoing bees.

Lines 139-140: Not clear, kindly re-phrase the sentence.

Line 213: number of honey bees’ incoming actions--- number of incoming honey bees

Line 21: number of honey bees’ outgoing actions.--- number of outgoing honey bees

Line 217: incoming action--- incoming bee

Line 218: outgoing action--- outgoing bee,

Lines 220-224: Not clear; kindly rephrase the sentences

Results

Good presentation, Kindly make following change.

Line 314: outgoing actions---outgoing bees 

Discussion: Appropriate

Conclusion: Satisfactory

References: Appropriate

Author Response

Dear reviewer:

We would like to thank you for your comprehensive revisions and valuable comments on our submitted manuscript entitled “A honey bee in-and-out counting method based on multiple object tracking algorithm”. The responses are highlighted in red color, and the amended changes are highlighted in blue color. We look forward to working with you to move this manuscript closer to publication in Insects.

Comments 1: I differ a little with the repeated arguments of the authors regarding the usefulness of this method in determining the colony health, refer: lines 10, 19, 29 etc. I suggest the authors to making this statement little general, as ‘usefulness of method in determining the colony condition’.

Response 1: Thank you for pointing this out. We agree with this comment. Therefore, we have changed ‘colony health’ to ‘colony condition’ in lines 10, 19, 21, 31 etc.

Comments 2: Title: A better title can be, “A Multiple Object Tracking Algorithm Method for Counting the In-and-Out Numbers of Honey Bees”

Response 2: In this article in-and-out counting methods were designed based on MOT, and our contribution lies in the method to count the bee’s in-and-out number. The current title is more suitable for our manuscript, and thank you for your advice on the title.

Comments 3: Line 22: after the word inefficient, add two sentences, ‘Therefore, there was a need to devise some more efficient alternate method. With this purpose the present study was conducted’.

Line 25: What type of camera was used? Kindly write.

Response 3: Agree. We have added two sentences in Line 22. We have inserted type of camara in Line 25 (now Line 26)

Line 22-23: Therefore, there was a need to devise some more efficient alternate method. With this purpose the present study was conducted.

Line 26: video stream data were captured using the camera (Huiboshi X20) on the smart beehive.

Comments 4: Introduction:

Lines 45-46: The number of bees in a colony— The condition of a colony

Line 46: outgoing counts---outgoing bees

Line 50: Counting bee in-and-out activity---Counting the numbers of in-and-out bees.

Line 90: Kindly check the sentence.

Line 96: outgoing actions---outgoing bees

Lines 97-100: These are redundant, kindly delete.

Response 4: Agree. We have modified the above issues.

Lines 45-46 (now Line 47): The number of bees in a colony--- The condition of a colony

Line 46 (now Line 48): outgoing counts---outgoing bees

Line 50 (now Line 52): Counting bee in-and-out activity---Counting the numbers of in-and-out bees.

Line 90 (now Lines 95-96): The current studies focus more on applicability of bee detection and in-and-out tracking.

Line 96 (now Line 101): outgoing actions---outgoing bees

Lines 97-100: we have deleted the sentence.

Comments 5: Materials and Methods

Looks sound. Kindly make following changes:

Line 109: to obtain the number of bees counts of incoming and outgoing actions—change to--- to count the number of incoming and outgoing bees.

Lines 139-140: Not clear, kindly re-phrase the sentence.

Line 213: number of honey bees’ incoming actions--- number of incoming honey bees

Line 214: number of honey bees’ outgoing actions--- number of outgoing honey bees

Line 217: incoming action--- incoming bee

Line 218: outgoing action--- outgoing bee,

Lines 220-224: Not clear; kindly rephrase the sentences

Response 5: Agree. We have modified the above issues.

Line 109 (now 111): to obtain the number of bees counts of incoming and outgoing actions—change to--- to count the number of incoming and outgoing bees.

Line 213 (now 221): number of honey bees’ incoming actions--- number of incoming honey bees

Line 214 (now 222): number of honey bees’ outgoing actions--- number of outgoing honey bees

Line 217 (now 225): incoming action--- incoming bee

Line 218 (now 226): outgoing action--- outgoing bee

Lines 220-224 (now 226-233): If a bee enters the detection box and exits immediately, the In-count and Out-count will both plus 1, which results in error. And the variable (Already-counted) is a double-ended queue and can solve the mentioned error. Already-counted is used to record the honey bee’s IDs which has entered the detection box. If a bee has exited the detection box but its ID has been recorded in Already-counted, the variable will remove the ID and the In-count will minus 1. After accessing all the objects in a frame, the next frame will be accessing until finishing all the frames of a video. And output the number of incoming and outgoing bees.

Comments 6: Results

Line 314: outgoing actions---outgoing bees

Response 6: Agree. We have modified the above issue.

Line 314 (now 336): outgoing actions---outgoing bees

Round 2

Reviewer 1 Report

Comments and Suggestions for Authors

none

Author Response

Response to Reviewer’ comments

Dear reviewer:

We would like to thank you for your comprehensive revisions and valuable comments on our submitted manuscript entitled “A honey bee in-and-out counting method based on multiple object tracking algorithm”. We look forward to working with you to move this manuscript closer to publication in Insects.

Sincerely yours,

Chaokai Lei

Reviewer 2 Report

Comments and Suggestions for Authors

The authors addressed all my questions.

Author Response

(The authors gave the same response as above.)
